# Moving morphable component (MMC) topology optimization with different void structure scaling factors

Zhao Li[1], Hongyu Xu[1]*, Shuai Zhang[2]

1 School of Mechatronics Engineering, Henan University of Science and Technology, Luoyang, China,
2 School of Vehicle and Traffic Engineering, Henan University of Science and Technology, Luoyang, China

* xuhongyu@haust.edu.cn

**Data Availability Statement:** All relevant data are within the paper and its Supporting Information files.

**Funding:** The financial supports from the Major Science and Technology Project of Henan Province

## Abstract

The explicit topology optimization method based on moving morphable component (MMC) has attracted more and more attention, and components are the basic building blocks of the implementation of MMC method. In the present work, a MMC topology optimization method based on component with void structure is followed with interest. On the basis of analyzing the characteristics of components used by MMC method, the topology description function for component with void structure is presented, where a quantitative scaling factor is introduced without increasing the number of design variables. Taking the minimum flexibility as the optimization objective, an example of short beam is analyzed with different void structure scaling factors. The results show that different scaling factors have a greater impact on the final topology optimization structure, and an ideal topology structure can be obtained with an appropriate scaling factor. Finally, some problems in the optimization process are analyzed and indicate that appropriate mesh density should be chose for component with void structure in order to achieve good optimization results.

## 1. Introduction

Topology optimization is the process of finding a reasonable layout of materials in the design domain to obtain the optimal structure response. Since the pioneering work of Bendsøe and Kikuchi [1], topology optimization methods have been widely developed. According to the expression of optimization result boundary, topology optimization methods can be divided into implicit topology methods, mainly including SIMP (Solid Isotropic Material with Penalization) method [2], evolutionary method [3, 4], and level set method [5–8], and explicit optimization methods, mainly including geometric mapping method [9], moving morphable component (MMC) method [10], and moving morphable void (MMV) method [11]. The boundary of implicit topology optimization result is often expressed in the form of pixels or nodes, which geometric information of the boundary cannot be directly obtained. A feasible design can be obtained only after complex post-processing by engineers with rich design experience. Compared with implicit topology optimization method, explicit topology optimization

(no.221100240400), and National Natural Science Foundation of China (no.51975244) are gratefully. acknowledged. The funders had no role in study design, data collection and analysis, decision to publish, or preparation of the manuscript.

**Competing interests:** The authors have declared that no competing interests exist.

method takes high-dimensional geometric related parameters as optimization design variables, which has clearer boundary expression. The result of explicit topology optimization can be seen as the optimal solution of the optimization problem and can be directly used by the CAD system [12–14], without errors in geometric information expression. Moreover, explicit topology optimization can also avoid some problems in implicit topology optimization, such as numerical instability, mesh-dependency, and dimensional disaster.

At present, MMC method has been one of the most active explicit topology optimization methods. Compared with traditional topology optimization methods where structural topologies are represented either by element densities (in density method) or by nodal values of a level set function (in level set method), in MMC method, a set of moving morphable components are adopted as basic building blocks of topology optimization. These components are allowed to move, deform, overlap and merge in the design domain freely, and structural topology can be obtained by changing the positions, inclined angles, lengths and the layout of these components. Moreover, these components comprise explicit geometric parameter information, which can be directly evolved to optimize structural topology.

MMC method takes the geometric information of components as design variables, which can accurately express the spatial position and the scale of components. First, topology optimization (TO) model should be created, mainly referring to the problem formulation, including design variable, objective function, and constraint. Then numerical implementation method should be established. In MMC framework, the number, type, and distribution of initial components should be determined firstly. A set of given initial layout components are projected onto the background mesh to obtain the information of mesh nodes in the projection area. The finite element analysis is used to solve the objective function. The design variables are updated based on constraint conditions and optimization algorithms, and the geometric information of the components will be changed. The optimal solution is obtained according to the convergence conditions. The implementation process of MMC method is shown in Fig 1.

From the implementation process of MMC method, it can be seen that appropriate component expression is the prerequisite for the implementation of MMC method and for improving the applicability of MMC method. In works [10, 11], components with straight skeletons (central lines) have been considered, as shown in Fig 2(A). After that, Guo et al. [15] proposed other structural components with curved skeleton explicitly in an elegant way, as shown in Fig 2(B). In their early work, the changes of component shape are mainly achieved by taking different control functions $f(x')$.

The description of components could affect the final topology optimization result, especially for some complex and subtle structures. Therefore, it is very important to develop more flexible components that are used for building blocks of optimization. There has been extensive research on component expression. Among them, it is widely studied to use different types of curves for component description. Zhang et al. [16] developed a topology optimization framework, which uses B-spline curves to describe the boundaries of moving morphable components in order to obtain the shape and topology of the structure simultaneously. Zheng and Kim [17] also combined the Non-uniform rational B-spline (NURBS) curves and traditional moving morphable components to create a new kind of morphable components. In this work, various kinds of complicated curved components can be built with NURBS curves or surfaces. Li et al. [18] demonstrated a new topology optimization approach derived from the MMC framework by replacing the straight components with the curved ones so as to enhance the geometric arbitrariness. The skeleton of the modified component is described by the NURBS curve, and the concept of time series is then proposed to directly generate the curved component from the 1D skeleton curve. On the basis of previous study [19], Liu et al. [20] developed B-spline curves as basic shape primitives for shape reconstruction and topology

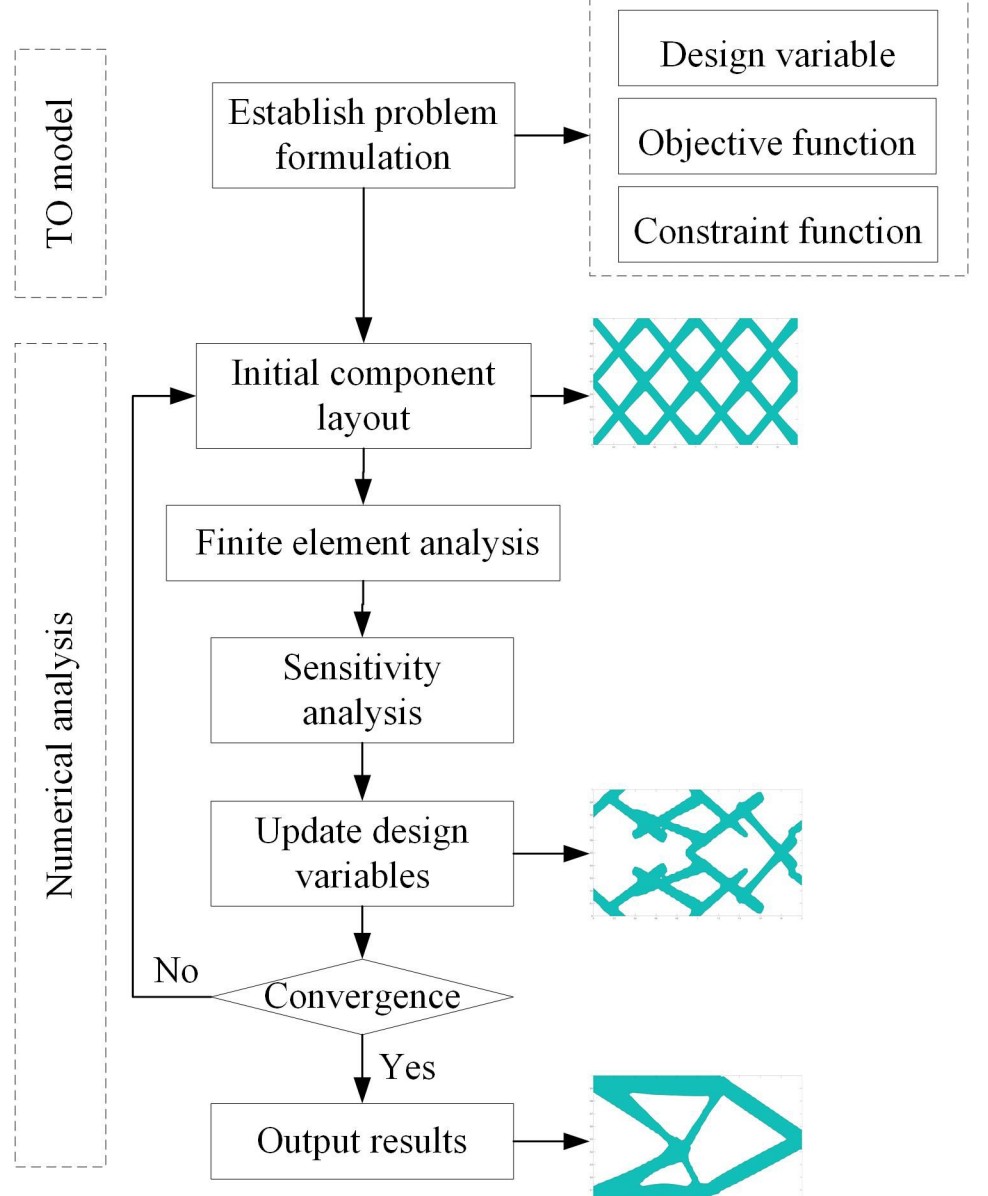

**Fig 1. The implementation process of MMC method.**

optimization. Further, Liu and Du [21] reconstructed the multiphase conductivity distributions in electrical impedance tomography (EIT) by MMC method. In this work, a signed distance-based shape topology description function is used to replace the hyperelliptic STDF topology description function in the traditional MMC approach. Besides B-splines curve, Bezier curve is also used for component description. Zhu et al. [22] presented a structural topology optimization method using moving wide Bezier components with constrained ends. The control points of wide Bezier curves are taken as design variables, and in order to form one single connected loadbearing structure, the loading, supporting, and/or some other functional interactions are connected by constraining the ends of wide Bezier curves. Shannon et al. [23] also introduced the generalized Bezier component into MMC-based framework. The

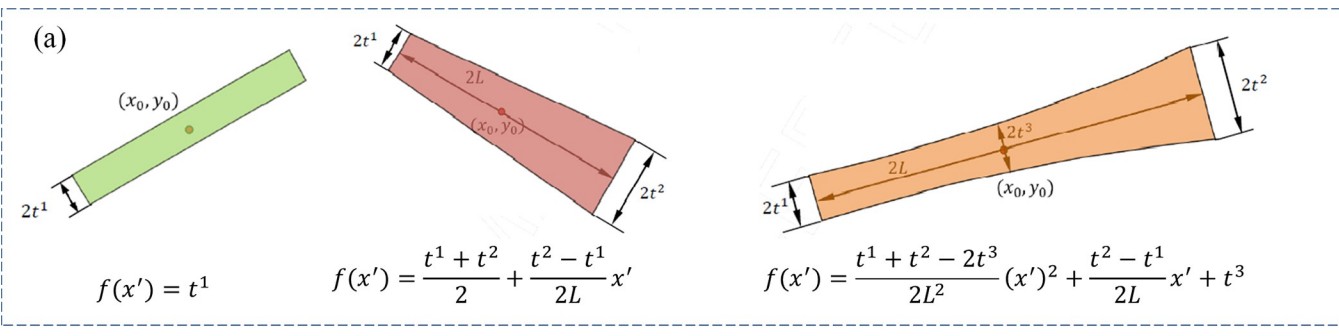

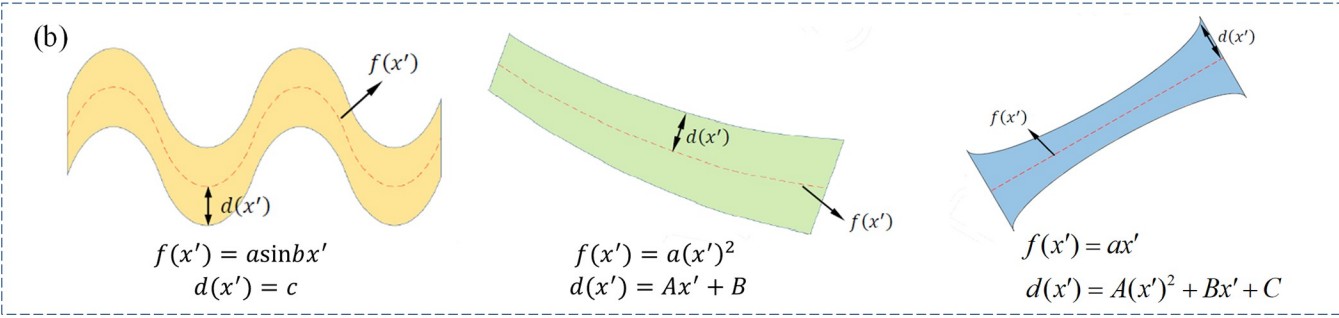

**Fig 2. The component shapes with different control functions ($f(x')$ and $d(x')$) determine the shape of the central line and the thickness variation respectively).** (a) The thickness variation is driven by $f(x')$ [11]. (b) The thickness variation is driven by $f(x')$ and $d(x')$ [15].

use of control point curves to represent structural components can provide additional flexibility in the shape and smooth curvature in structural boundaries.

There are also other component description forms. Deng and Chen [24] designed a novel connected morphable component (CMC). In this work, the components, which two ends are circulars, are connected through shared end points to prevent the degradation of the flexible structure and ensure the integrity/connectivity of the structure. Wang et al. [25] described the geometric shape of components with hinge characteristic, supposing the central part of the component with profile described by second-order polynomial is replaced by corner-filleted hinge, which is composed of thin straight-beam and transition fillets. Based on the descriptions of areas, Yang and Huang [26] discussed the geometric descriptions of different components, and an extended form of the multi-section component was constructed to improve the deformation ability of components. To solve the shortage of C$^1$ continuity, Otsuka et al. [27] proposed an adaptive moving morphable component (ANCF) component with two nodes based on absolute nodal coordinate formulation. Because both the position and gradient are used as design variables, C$^1$ continuity is ensured.

Previous studies on component expression have mostly focused on controlling the change of the peripheral boundary of components, in order to obtain more smooth or detailed component changes. However, few scholars paid attention to the changes in the internal structure of components. Guo et al. [10] proposed that geometric features can be embedded in components for structural optimization, including void structures. Zhang et al. [28] proposed a sandwich structure component, where the internal and external structures of the component are made of different materials. The material properties of the internal structure are controlled by limiting the intersection area of the components. When the internal sandwich structure material is empty, it can be regarded as a component with void structure. Bai et al. [29] proposed a three-dimensional topology optimization approach to obtain the hollow structures by using moving morphable components, which the hollow component can be described by internal

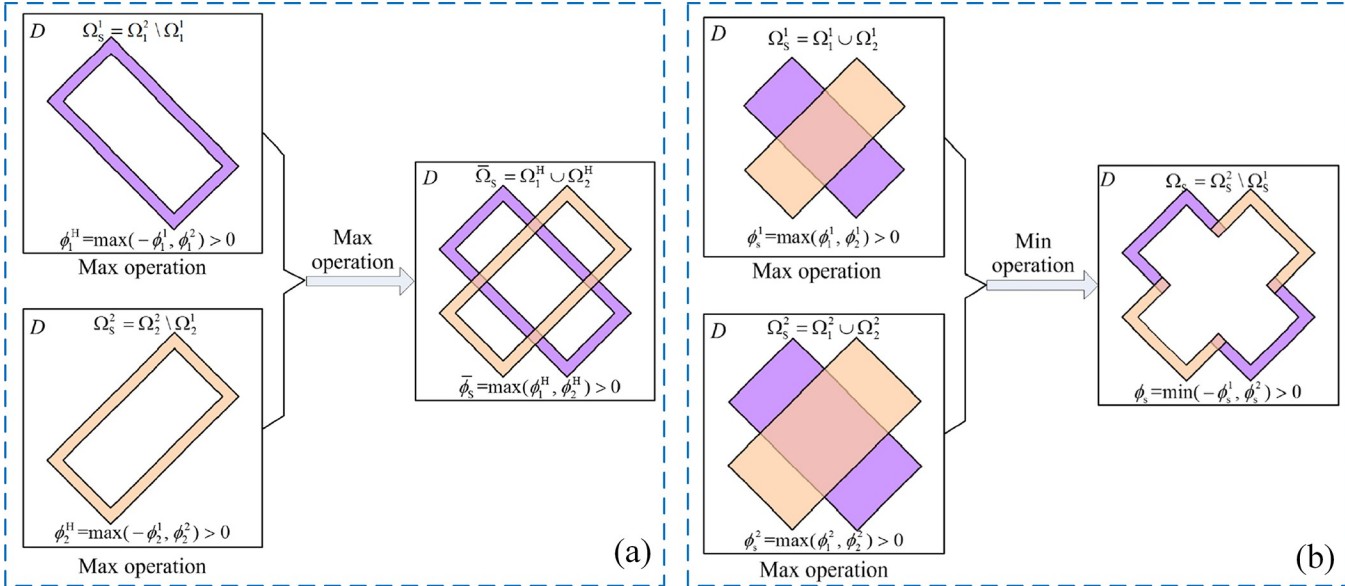

**Fig 3. Construction strategy for hollow structure component [29].** (a) Construct hollow structure by the operation of Boolean "AND". (b) Construct hollow structure by the operation of Boolean "AND" and "NOT" between internal and external solid components.

and external components topology description functions respectively. In this work, two construction strategies for hollow structure component are proposed, as shown in Fig 3. However, the previous studies usually used predetermined fixed scale internal void structure.

In the present work, we focus on the impact of internal void scale changes on topology optimization results. The rest of the paper is organized as follows. In section 2, a detailed introduction to the characteristics of the components on the MMC method is provided. The description functions of components with void structures are introduced in Section 3. The problem description function is provided in Section 4. The impacts of different void structure scaling factors on the optimization results and some numerical problems are discussed in Section 5.

## 2. Components in MMC method

### 2.1 Principle of components evolution

Different from density method driving artificial densities and level set method driving nodes, MMC method mainly changes the topology structure by driving the geometric parameters of the components. In MMC method, components have clear parameterized geometric expressions. As long as the position and scale parameters of a certain number of components in the design domain are determined, the topology structure can be explicitly determined. In terms of components evolution, the translation and rotation of components are realized by changing the center points coordinates and rotation angles of components. The shape of components changes through the updates of components scale information. It should be noted that the initial layout quantity of components is generally controlled by humans, and the extinction of components is realized through overlapping components.

### 2.2 Explicit expression characteristics of components

The simplest component can be described by 5 design variables, including the coordinates of component center point, component thickness, component width, and rotation angle, as

shown in Fig 2(A). If every mesh in the density method or every node in the level set method is assumed to be a "component", the driving variables of "component" are smaller than those in MMC method. But the number of divided meshes and nodes in the same design domain is much greater than that of components, so the total number of design variables is much greater than that of MMC method. The smaller total number of variables determines the MMC method has higher optimization efficiency. In terms of optimization accuracy, the boundary obtained by density method is zigzag, and the boundary obtained by level set method is some discrete nodes, which both require later fitting to obtain the smooth structural topology. The fitted boundary is only an approximate boundary. MMC method directly drives components with explicit geometric parameters to perform boundary evolution, which will be mapped onto the fixed meshes of design domain to get finite analysis model. The topology optimization results directly correspond to the geometric information of components. From the perspective of building a structure model, the same structure can be represented by an accurate geometric modeling or an approximate finite element model respectively. The idea of implicit topology optimization method can be understood as building a finite element model first, then driving the finite element model to obtain the optimal structure, and finally obtaining the approximate solution of the geometric modeling. The idea of MMC method can be understood as building a geometric modeling first, then establishing a finite element model from the geometric modeling, through driving the geometric modeling to constantly change the finite element model to get the optimal solution, and finally getting the direct solution of the geometric modeling, which is the explicit expression principle of components.

## 2.3 Finite element analysis based on components

In MMC method, components are mapped onto the background meshes, so the components and background meshes are completely decoupled theoretically. When finite element analysis is carried out, only the nodes in the coverage area are the effective analysis nodes. Therefore, the degree of freedom (DOF) reduction techniques can be adopted to further improve the analysis efficiency [28, 30]. Moreover, it is necessary to process the nodes at the component boundary to ensure the continuity of finite element analysis. The relationship between the components and the background meshes is shown in Fig 4(A). The component boundary will cut the edge of the background mesh, which means that the complete mesh may be not included at the boundary, as shown in Fig 4(B). To ensure calculation accuracy, appropriate processing technology should be done.

At present, there are two commonly used finite element analysis methods in the MMC method: extended finite element method (X-FEM) and the use of ersatz material model. The X-FEM is to reconstruct the mesh near the structure boundary by linear interpolation according to the values of the topological describing function at the element node to improve the calculation accuracy near the boundary, as shown in Fig 4(C). However, the X-FEM increases the number of nodes, which inevitably increases the computation. Using ersatz material model, the stiffness of the element is connected with the values of the topological describing function on its nodes, and the Young's model of the relevant area is recalculated through the Heaviside function, as shown in Fig 4(D). In present work, ersatz material model method is used for finite element analysis.

## 3. Description of components with void structure

In order to facilitate the analysis of the characteristics of components with void structure, the equal thickness rectangular components described by the hyperelliptic function is adopted as the basic research object.

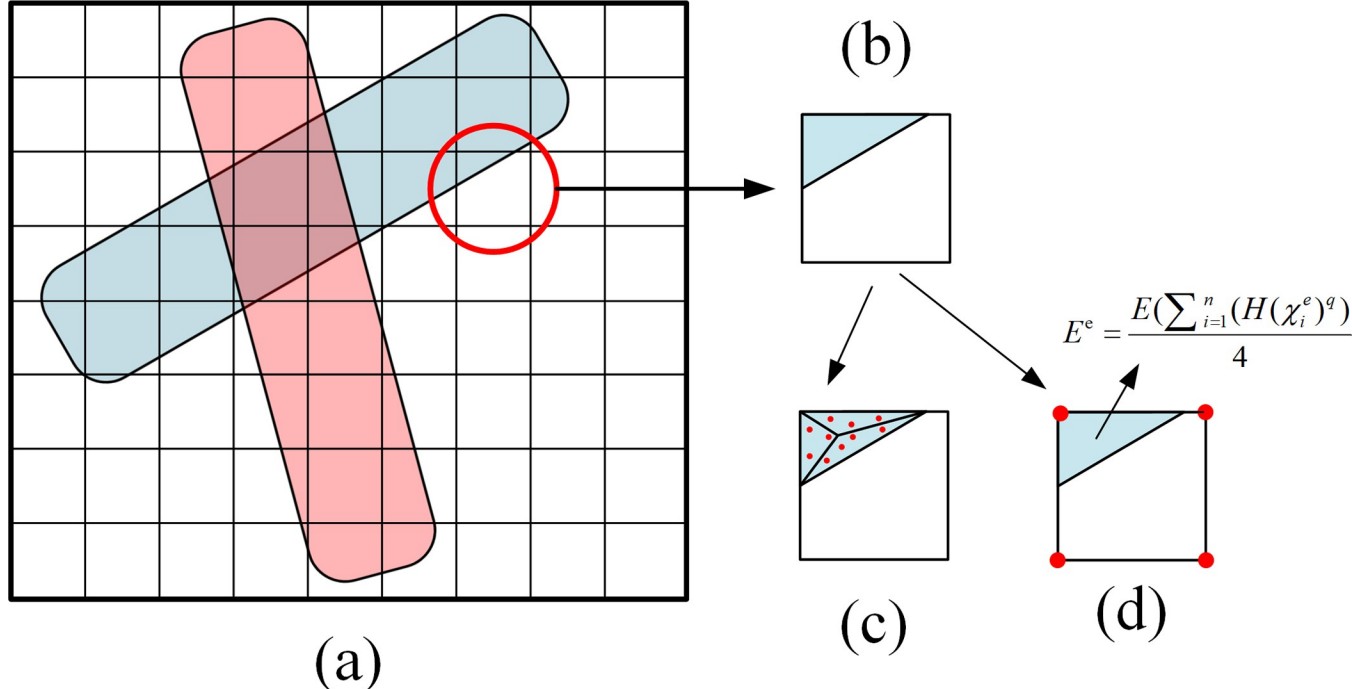

**Fig 4. Finite element analysis based on component.** (a) Schematic diagram of the relationship between components and background meshes. (b) Schematic diagram of component boundary cutting meshes. (c) Extended finite element method (X-FEM). (d) Ersatz material model method.

### 3.1 Basic component description

The schematic diagram of component geometric parameters is shown in Fig 5. The description and evolution of components are the core of MMC method, and the final topology optimization structure is also determined by component description variables. At present, the most

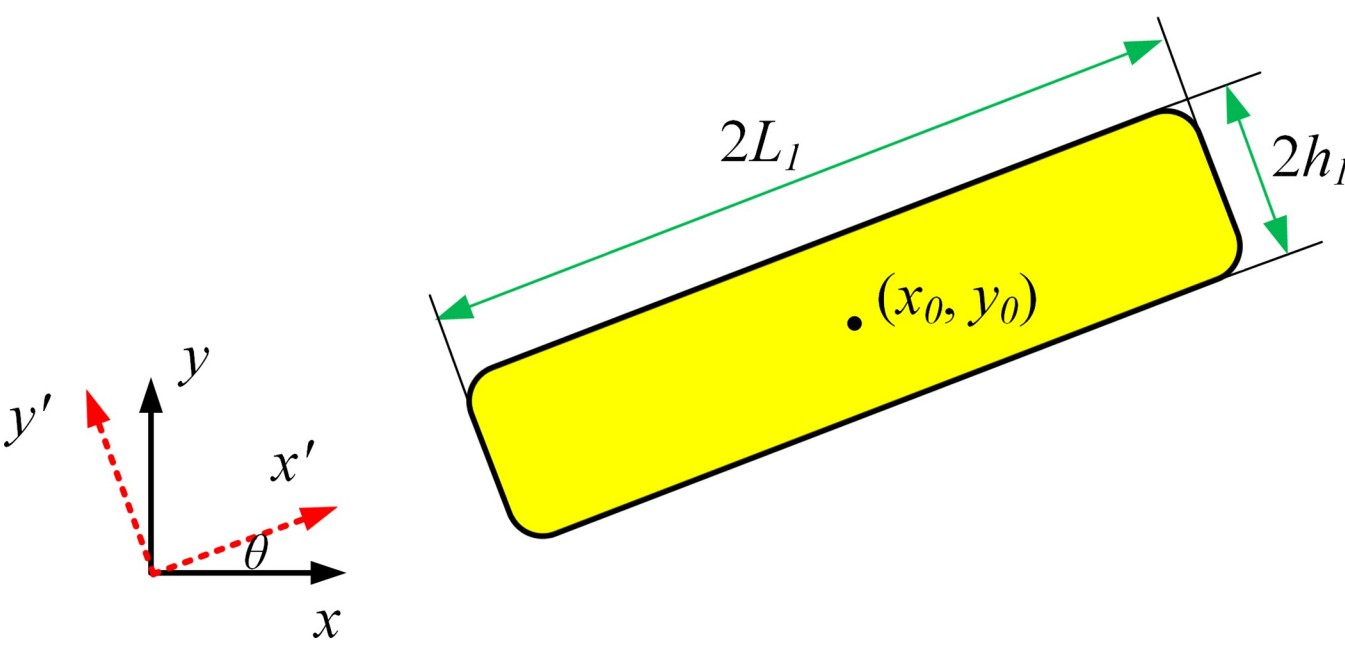

**Fig 5. Basic component.**

commonly used component description method is Euler description, which describes components based on their position and scale information. Components can be described by the following hyperelliptic function:

$$\phi(x,y) = 1 - \left(\frac{x'}{L}\right)^p - \left(\frac{y'}{f(x')}\right)^p \tag{1}$$

with

$$\begin{Bmatrix} x' \\ y' \end{Bmatrix} = \begin{bmatrix} \cos\theta & \sin\theta \\ -\sin\theta & \cos\theta \end{bmatrix} \begin{Bmatrix} x - x_0 \\ y - y_0 \end{Bmatrix} \tag{2}$$

where $\phi(x,y)$ represents the topological function value of the coordinates corresponding to the entire design domain $D$. When $\phi(x,y) = 0$, it describes the component boundary. The parameter $p$ is a relatively large even integer number (usually take $p = 6$). $L$ is the half length of the component, $f(x')$ is a function describing the shape of the component. $(x',y')$ and$(x, y)$represent the coordinate positions of any point in the local and global coordinate systems respectively. $\theta$ is the rotation angle of the component (the rotation angle from the global coordinate system to the local coordinate system). However, it should be noted that pure Lagrangian geometry description is also applicable to develop MMC topology optimization method, and relevant research can be referred to [10, 28, 29].

After determining the component description function and the number of components, the entire design domain can be divided into the following areas with the components:

$$\begin{cases} \phi^S(x) > 0, & if\ x \in \Omega^S \\ \phi^S(x) = 0, & if\ x \in \partial\Omega^S \\ \phi^S(x) < 0, & if\ x \in D\backslash(\Omega^S \cup \partial\Omega^S) \end{cases} \tag{3}$$

where $\Omega^s$ is the area occupied by solid materials in the design domain, that is, $\phi^S(x) > 0$ and $\phi^S(x) = 0$ are the reserved areas for topology optimization materials. The topology describing function of all components can be expressed as:

$$\phi^S(x) = \max(\phi_1^e, \cdots, \phi_i^e, \cdots, \phi_n^e) \tag{4}$$

where $n$ represents the number of components. $\phi_i^e$ is the area occupied by $i$-th component, which can be determined by the following function:

$$\begin{cases} \phi_i^e(x) > 0, & if\ x \in \Omega^i \\ \phi_i^e(x) = 0, & if\ x \in \partial\Omega^i \\ \phi_i^e(x) < 0, & if\ x \in D\backslash(\Omega^i \cup \partial\Omega^i) \end{cases} \tag{5}$$

where $\Omega^i$ is the area occupied by the solid material in the $i$-th component.

### 3.2 Component description with void structure

The schematic diagram of component geometric parameters with void structure is shown in Fig 6. The component can be descripted as $\phi_{ij}^e$ ($i = 1,2,\ldots,$n; $j = 1,2$), where $\phi_{i1}^e$ represents solid material area and $\phi_{i2}^e$ represents void area. The corresponding description function can be written as:

$$\phi_{i1}^e(x,y) = 1 - \left(\frac{x'}{L_1}\right)^p - \left(\frac{y'}{h_1}\right)^p \tag{6}$$

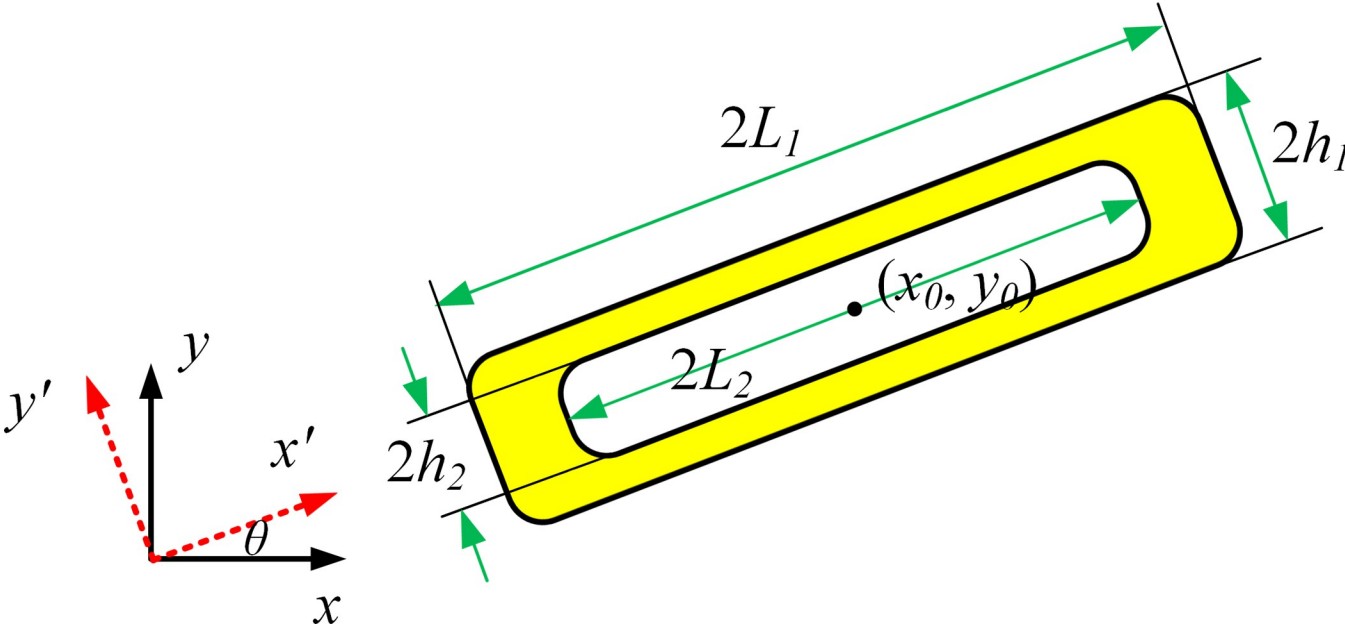

**Fig 6. Component with void structure.**

$$\phi_{i2}^e(x,y) = 1 - \left(\frac{x'}{L_2}\right)^p - \left(\frac{y'}{h_2}\right)^p \qquad (7)$$

where $L_1$, $L_2$, $h_1$, $h_2$ have the following relationships:

$$\begin{cases} L_2 = a \cdot L_1 \\ h_2 = b \cdot h_1 \end{cases} \qquad (8)$$

where $a$ is the length direction scaling factor and $b$ is the width direction scaling factor.

### 3.3 Distribution description of topological function values

Distribution of topological function values for component with void structure is shown in Fig 7. Because the internal and external structures of component are described separately, the distribution description of topological function values within the design domain has the following forms:

1. Topological description of the area where $A_1$ is located: $\phi_{i1}^e < 0$ and $\phi_{i2}^e < 0$.

2. Topological description of the area where $A_2$ is located: $\phi_{i1}^e > 0$ and $\phi_{i2}^e < 0$.

3. Topological description of the area where $A_3$ is located: $\phi_{i1}^e = 0$ or $\phi_{i2}^e = 0$.

4. Topological description of the area where $A_4$ is located: $\phi_{i1}^e > 0$ and $\phi_{i2}^e > 0$.

The ultimate goal of topology optimization is to obtain the information of all component areas $A_2$ and $A_3$. The essence of the component description function is that the solid area is positive and the void area is negative. For components with void structure, the topological function value of the unit component area can be obtained through the min function. Unit components $\phi_i^e$ consists of solid structure $\phi_{i1}^e$ and void structure $\phi_{i2}^e$, and the implementation process is shown in Table 1.

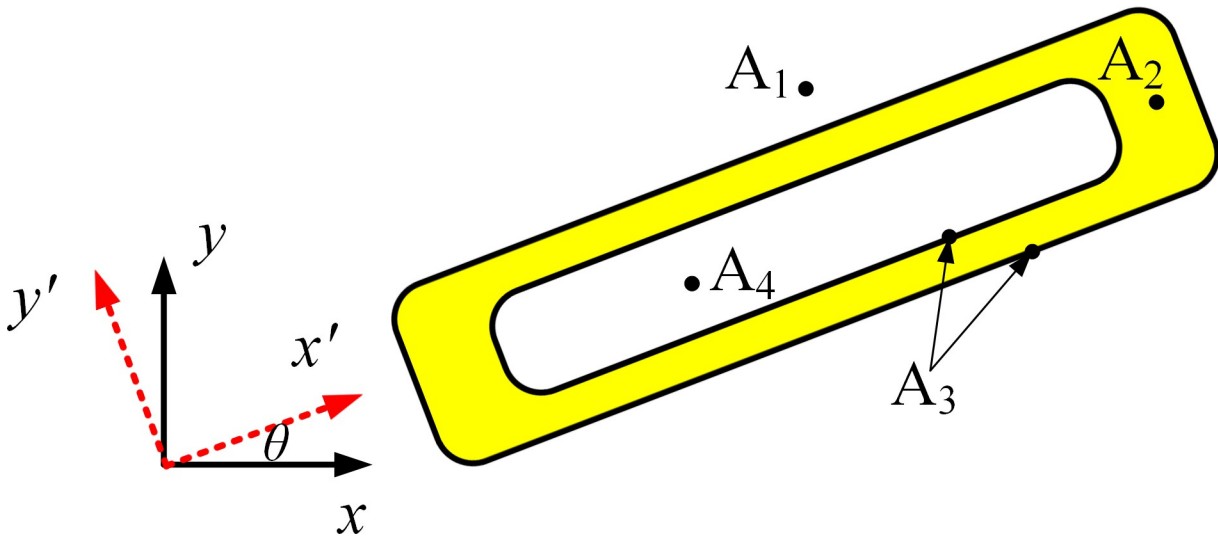

**Fig 7. Schematic diagram of topological function value area.**

Therefore, component with void structure can be described as:

$$\phi_i^e(x, y) = \min(\phi_{i1}^e(x, y), -\phi_{i2}^e(x, y)) \tag{9}$$

Then the topology function values of all component areas are obtained through the max function.

## 4. Problem function

### 4.1 Minimum flexibility problem

Taking volume as constraint and structural flexibility as objective function, the specific optimization formula is as follows:

$$Find \quad D = ((D^1)^T, \dots, (D^i)^T, \dots, (D^n)^T)^T, u(x)$$

$$Min \quad C = \int_D H(\phi^s(x, D)) f \cdot u dV + \int_{\Gamma_t} t \cdot u dS$$

$$S.t.$$

$$\int_D H(\phi^s(x, D)) E : \varepsilon(u) : \varepsilon(v) dV = \int_D H(\phi^s(x, D)) f \cdot v dV + \int_{\Gamma_t} t \cdot v dS, \forall v \in U_{ad} \tag{10}$$

$$V - \bar{V} = \int_D H(\phi^s(x, D)) dV - \bar{V} \leq 0$$

$$u = \bar{u}, \quad on \ \Gamma_u$$

$$D \subset U_D$$

**Table 1. Obtaining solid areas of components with void structure.**

| Topological function value | Area $A_4$ | Area $A_2$ | Area $A_1$ | Inner boundary | Outer boundary |
|---|---|---|---|---|---|
| $\phi_{i1}^e$ | + | + | - | + | 0 |
| $\phi_{i2}^e$ | + | - | - | 0 | - |
| $-\phi_{i2}^e$ | - | + | + | 0 | + |
| $\phi_i^e = \min(\phi_{i1}^e, -\phi_{i2}^e)$ | - | + | - | 0 | 0 |

where $D$ represents the structural design domain. $f$ and $t$ denote the body force density and the surface traction on the Newman boundary $\Gamma_t$, respectively. $u(x)$ represents the displacement field function of the structure in the design domain, which is only related to the coordinate $x$. $v(x)$ represents the trial displacement function, and $U_{ad} = \{v(x)|v(x) \in H^1(D), v(x) = 0 \ on \Gamma_U\}$ is the allowable set that can be selected for the displacement trial function. $\bar{u}$ is the prescribed displacement on Dirichlet boundary $\Gamma_U$. $E(x) = (E_{ijkl}(x))$ is the fourth order elastic tensor distributed within the design domain with $E = E/(1 + v)[\| + v/(1 - 2v)\delta \otimes \delta]$, where II is the fourth order unit tensor, $\delta$ is the second order unit tensor, and $\bigotimes$ is the tensor product. $H(x)$ represents the Heaviside function. Since the Heaviside function is not a continuous function, for the sake of sensitivity derivability and numerical stability of iterative optimization, the original discontinuous Heaviside function is often replaced by the continuous approximation function near $x = 0$. The form of $(x)$ is taken as:

$$H_\tau(x) = \begin{cases} 1, & x > \tau \\ \dfrac{3(1 - \alpha)}{4}\left(\dfrac{x}{\tau} - \dfrac{x^3}{3\tau^3}\right) + \dfrac{1 + \alpha}{2} & -\tau \leq x \leq \tau \\ \alpha, & x < -\tau \end{cases} \qquad (11)$$

where $\tau$ is a parameter that controls magnitude of regularization and $\alpha$ is a small positive number to ensure the nonsingular of the global stiffness matrix.

Structural flexibility $C$ represents the deformation energy of solid structure, which is the sum of deformation energy caused by body and surface forces. Based on finite element analysis ideas, structural flexibility $C$ is also equal to the sum of strain energy of elements. In MATLAB program implementation, finite element analysis is used to calculate the structural flexibility. And in MMC framework, the premise of finite element analysis is to obtain solid element information based on nodes' topological function values and Heaviside function The stiffness matrix is solved by ersatz material model. In our proposed method, the role of the scaling factors ($a$ and $b$) is mainly worked in program implementation. The voids generated by the scaling factors ($a$ and $b$) are used to remove the internal nodes of the voids. The removed internal nodes of the voids will no longer undergo finite element analysis, and the removal of voids is achieved by the *Min* function in this paper.

## 4.2 Sensitivity analysis

The sensitivity of the objective function to any design variable $a$ (component geometric parameters) is:

$$\frac{\partial C}{\partial a} = -u^T \frac{\partial K}{\partial a} u = -u^T\left(\frac{E}{4}\left(\sum_{e=1}^{NE} \sum_{i=1}^{4} q(H(\phi_i^e))^{q-1} \frac{\partial H(\phi_i^e)}{\partial a} k^s\right)\right)u \qquad (12)$$

where $K$ is the overall stiffness matrix of the structure. $k^s$ is the stiffness matrix of the elements corresponding to the topological function values $\phi_i^e = 1$, i = 1,..., 4, and $E = 1$. In(12), $NE$ is the total number of elements in the design domain.

The sensitivity of the constraint function, also known as the volume constraint function, to the design variable $a$ is:

$$\frac{\partial V}{\partial a} = \frac{1}{4}\sum_{e=1}^{NE} \sum_{i=1}^{4} \frac{\partial H(\phi_i^e)}{\partial a} \qquad (13)$$

## 5. Example analysis

### 5.1 Components with single void structure

In the present work, the short beam problem is taken as an example for analysis. The initial layout of the component with single void structure is shown in Fig 8. The units are all dimensionless, with the design domain (1×2), divided into 200×100 meshes. The displacement is set to zero along the left side of the design domain, and the loading is set at the midpoint of the right end. The initial layout consists of 16 components with void structure that intersect with each other. It should be noted that the evolution of components is achieved by the method of moving asymptotes (MMA) optimizer based on gradient information, and the initial symmetrical layout of components is only convenient for setting. When a stable load-transferring path is established, it is the truly effective "initial layout".

The optimization results of different scaling factors are shown in Fig 9. It can be seen that:

1. After introducing void structure, the topology expression form of the structure is enriched, and under some certain conditions, more suitable structural forms for manufacture can be obtained.

2. Different scaling factors have a greater impact on the final optimized structure. In this case, when the scaling factor $a = b = 0.5$, the topology optimization structure is better with manufacturable contour.

3. After introducing void structure, the minimum wall thickness of the component will be reduced, and scale limitations should be considered from both manufacture and calculation perspectives to improve the feasibility of optimizing structure. If the scaling factor in the width direction is too large (such as $b = 0.8$), the stiffness matrix is close to singularity, and it is difficult to obtain an accurate optimal solution, which also indicates that the scaling factor in the width direction is more sensitive to the optimization results.

4. If the scaling factor is too small, the impact on the topology structure is also small, which can be used as a reference for determining the void position of the original structure.

The layout of the components can be identified in Fig 10. It can be clearly seen that the lengths of components have changed and the components are overlapped at the same location.

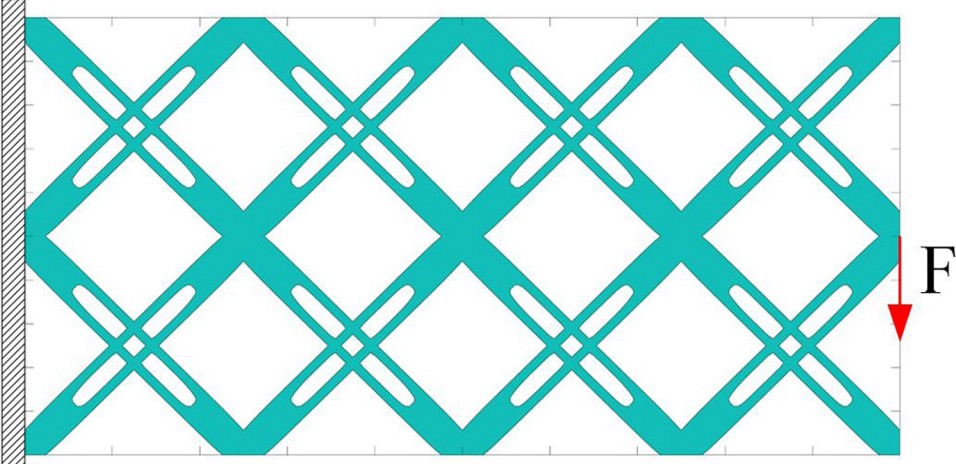

**Fig 8. Initial layout of components with single void structure.**

## 5.2 Components with double voids structure

Component with double void structures is shown in Fig 11. After setting the center distance to a fixed value, the coordinates of the void center point can be represented by the coordinates of the component center point without increasing the number of variables, as shown in Fig 11 (B). The conversion functions are as follows:

$$\begin{cases} x_1 = x_0 - \dfrac{L_1}{2} \bullet \cos\theta \\ y_1 = y_0 - \dfrac{L_1}{2} \bullet \sin\theta \end{cases} \quad \begin{cases} x_2 = x_0 + \dfrac{L_1}{2} \bullet \cos\theta \\ y_2 = y_0 + \dfrac{L_1}{2} \bullet \sin\theta \end{cases} \tag{14}$$

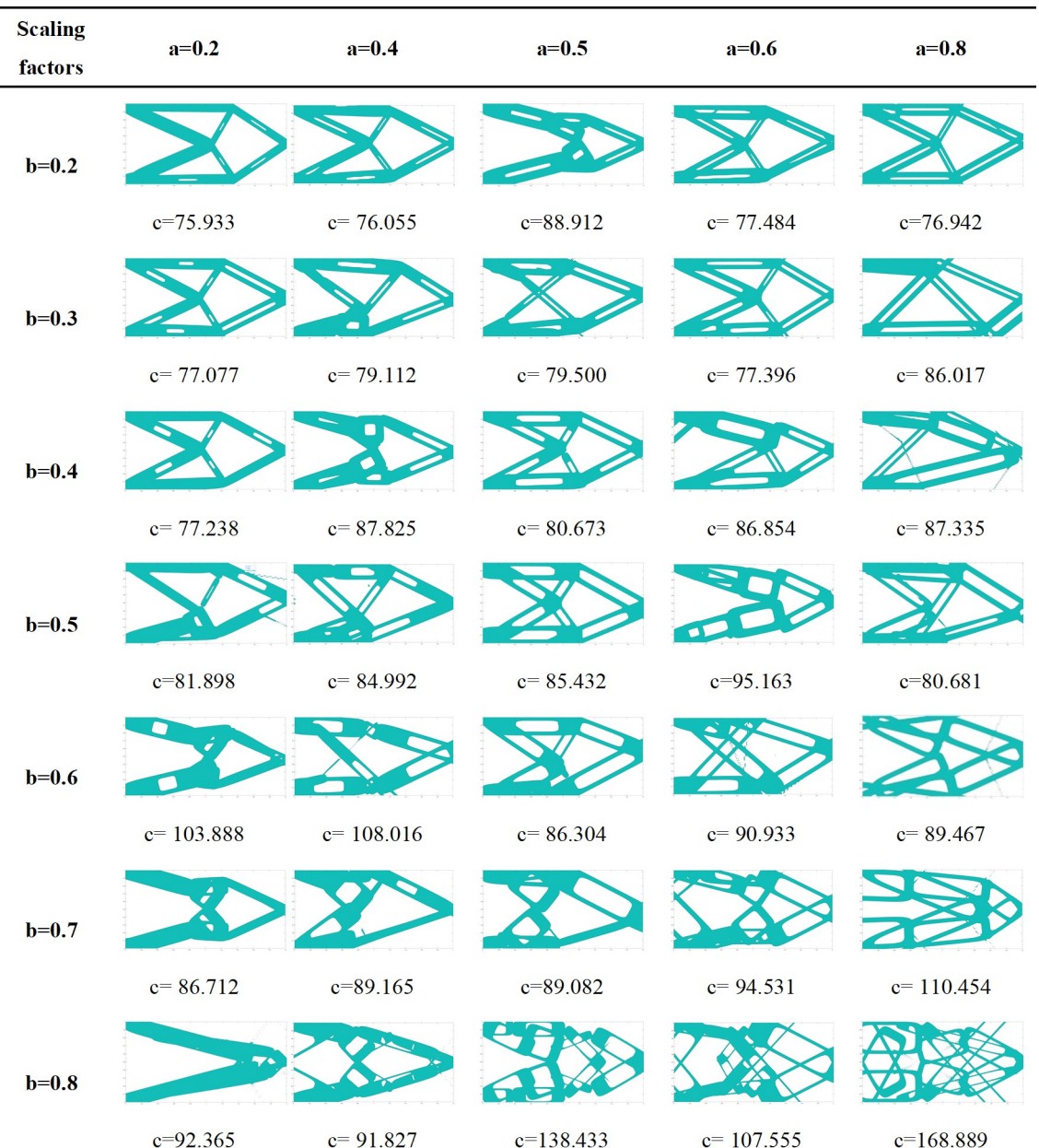

| Scaling factors | a=0.2 | a=0.4 | a=0.5 | a=0.6 | a=0.8 |
|---|---|---|---|---|---|
| b=0.2 | c=75.933 | c= 76.055 | c=88.912 | c= 77.484 | c=76.942 |
| b=0.3 | c= 77.077 | c= 79.112 | c= 79.500 | c= 77.396 | c= 86.017 |
| b=0.4 | c= 77.238 | c= 87.825 | c= 80.673 | c= 86.854 | c= 87.335 |
| b=0.5 | c=81.898 | c= 84.992 | c= 85.432 | c=95.163 | c=80.681 |
| b=0.6 | c= 103.888 | c= 108.016 | c= 86.304 | c= 90.933 | c= 89.467 |
| b=0.7 | c= 86.712 | c=89.165 | c=89.082 | c= 94.531 | c= 110.454 |
| b=0.8 | c=92.365 | c= 91.827 | c=138.433 | c= 107.555 | c=168.889 |

**Fig 9. Optimization results of different scaling factors.**

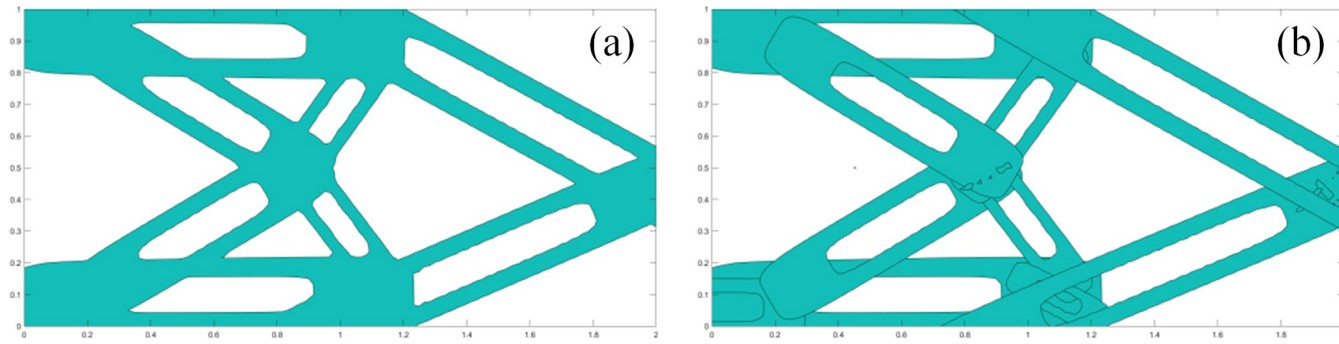

**Fig 10. Optimization result based on components with single void structure ($a = b = 0.5$).** (a) Contour plot. (b) Component plot.

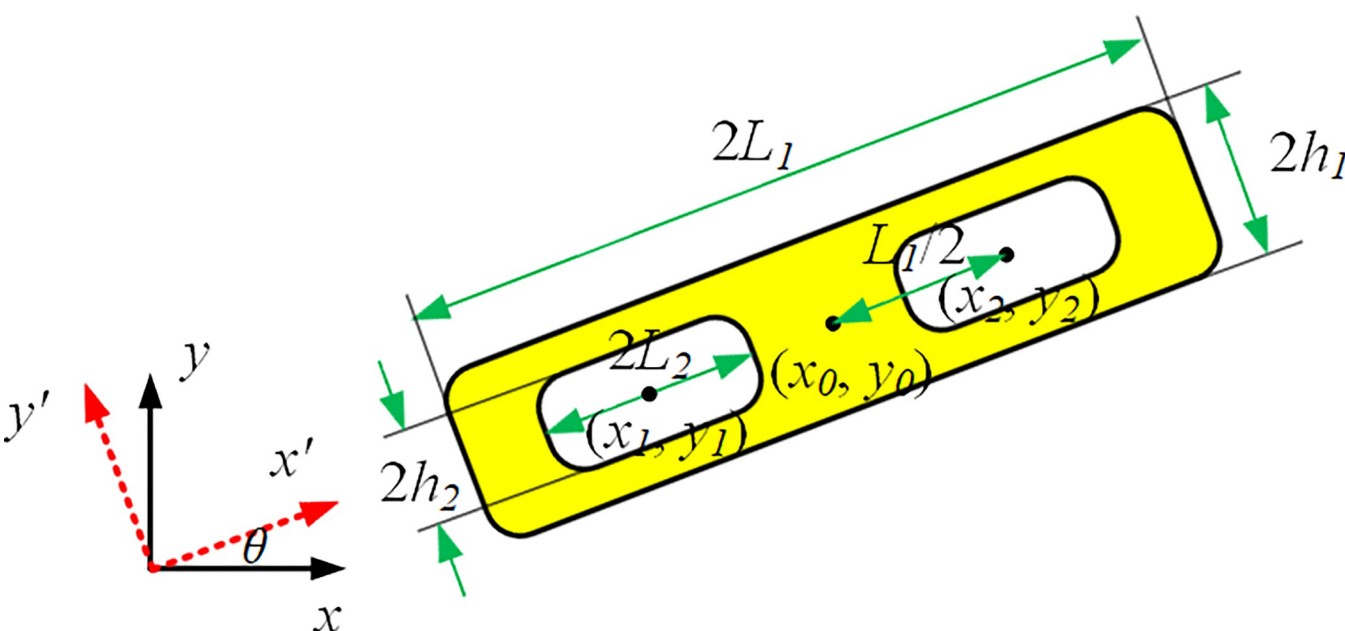

(a) The schematic diagram of component geometric parameters

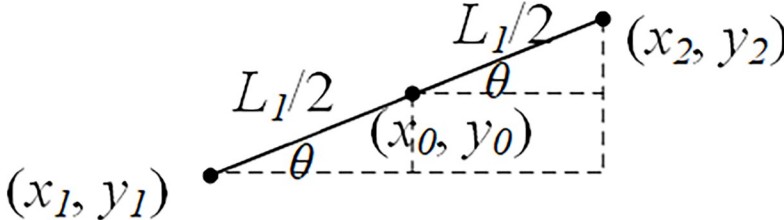

(b) Center distance relationship between double void structures

**Fig 11. Component with double void structures.**

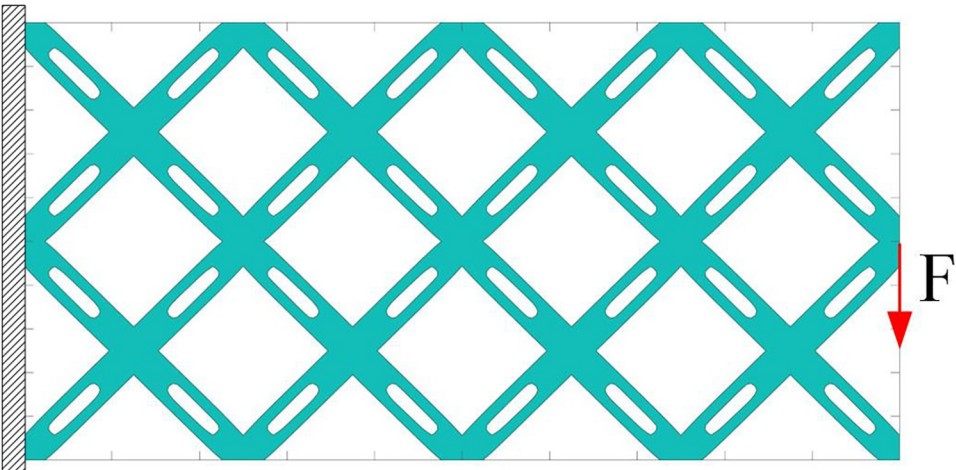

**Fig 12. Initial layout of components with double void structures.**

Two voids are set with the same size structure. The scaling reference for the width direction of the void is the half of component width, and the scaling reference for the length direction of the void is the half of component length. The scaling factor in the width direction is consistent with the component with single void structure, but the scaling factor in the length direction does not exceed 0.5. The initial layout of the component with double void structures is shown in Fig 12, and the boundary conditions are the same as example of single void structure. The optimization results of different scaling factors are shown in Fig 13. It can be seen that, after

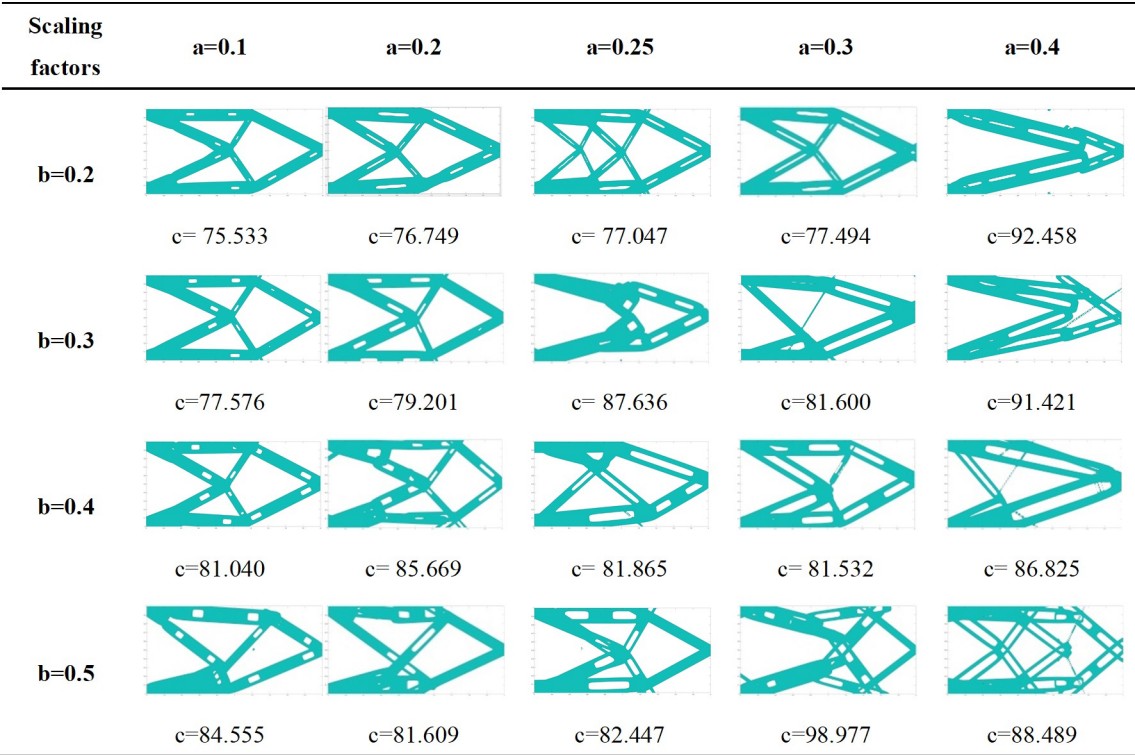

| Scaling factors | a=0.1 | a=0.2 | a=0.25 | a=0.3 | a=0.4 |
|---|---|---|---|---|---|
| b=0.2 | c= 75.533 | c=76.749 | c= 77.047 | c=77.494 | c=92.458 |
| b=0.3 | c=77.576 | c=79.201 | c= 87.636 | c=81.600 | c=91.421 |
| b=0.4 | c=81.040 | c= 85.669 | c= 81.865 | c= 81.532 | c= 86.825 |
| b=0.5 | c=84.555 | c=81.609 | c=82.447 | c=98.977 | c=88.489 |

**Fig 13. Optimization results of different scaling factors.**

adding voids, the computational efficiency will decrease. Moreover, when the total length of the initial layout voids is equal (that is the length of single void is equal to the sum of the lengths of two voids in double voids structure), there are also some differences in the final optimization results. The optimized structures of single void are smoother, which becomes more pronounced as the scaling factor increases. As the complexity of component structure increases, the detailed structure will also increase, making the optimized structures of double voids having more void expression. It is necessary to clarify whether those details are required. Comparing the computational efficiency and optimization results of single void and double voids structure of present study, single void structure for structural optimization is suggested to use.

### 5.3 Analysis of related problems

**5.3.1 Analysis of the same structural component with different description.** There exists a special case in component with void structure where the middle of component is completely penetrated. The component can be described in different forms by single void or double voids. In single void structure, setting $a = 1$ can describe hollow component. In double voids structure, setting $a = 0.5$ can describe hollow component. However, the optimization results are not entirely consistent. As shown in Fig 14, the optimization results of component with single void structure is better, and as the width scaling factor increases, the differences between single void and double voids become greater.

**5.3.2 Analysis of the optimization accuracy.** The void structure makes the component thinner, which has significant impact on the final optimization result, including the topology and minimum flexibility value. Some void structures even cause the optimization results without practical significance. However, compared with the optimization results of solid component, as shown in Fig 15, the void structure with an appropriate scaling factor can achieve the similar accuracy.

Take $a = b = 0.5$ and change the loading position, the topology optimization results are shown in Fig 16. The optimization results indicate that a good topology structure can also be obtained under different loading positions by using components with void structure.

**5.3.3 Numerical problem.** The parameterized geometric modeling in MMC method can be independent of the analysis mesh, so different numerical analysis methods can be used to calculate the structural response. However, when using the finite element method to analyze the structural response of the geometric modeling, it needs to be converted into a finite

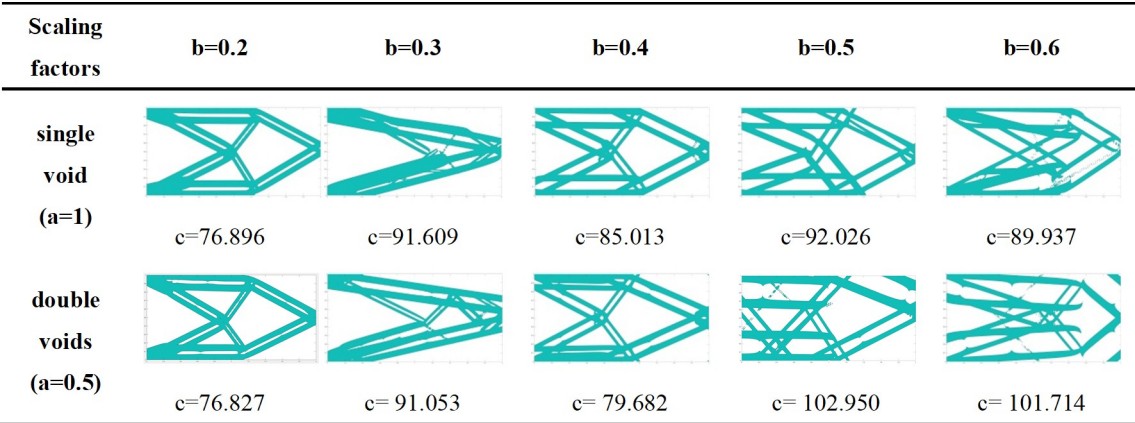

| Scaling factors | b=0.2 | b=0.3 | b=0.4 | b=0.5 | b=0.6 |
|---|---|---|---|---|---|
| single void (a=1) | c=76.896 | c=91.609 | c=85.013 | c=92.026 | c=89.937 |
| double voids (a=0.5) | c=76.827 | c= 91.053 | c= 79.682 | c= 102.950 | c= 101.714 |

**Fig 14. Optimization results based on the same structural component with different description.**

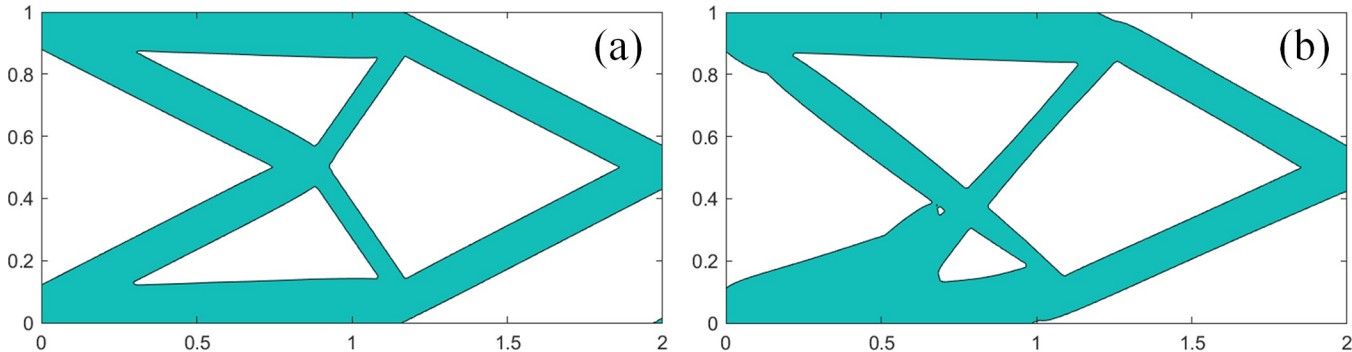

**Fig 15. The optimization result of solid component.** (a) Optimization result based on uniform thickness component(c = 74.569). (b) Optimization result based on linearly varying thickness component (c = 76.785).

element analysis model. So there are also some numerical problems in topology optimization under the MMC framework. One of them is the "islands" phenomenon. In the process of optimization, some component structures are too small to be captured by the mesh (relatively speaking, it can also be due to the excessive mesh division scale). These structures will not work in finite element analysis, but will still display in the final optimization result, independent of the main structure. For components with void structure, the "islands" phenomenon is more easily to occur due to their relatively smaller local dimensions.

According to the topology description function of component, $\Phi \geq 0$ is the solid material area, where the values of $\Phi$ are determined by the node coordinates (the function values at the node). If the local structure of the component is too small or the background mesh is too large, it cannot effectively cover all node information, resulting in discontinuous expression of the component, that is the "island" phenomenon, as shown in Fig 17. The "island" will affect the accuracy and convergence of topology optimization, and especially should be avoided during the initial layout of components. Scale limitations and appropriate mesh density can effectively solve "island" numerical problem.

In addition, there are still some limitations of proposed method at present. In manufacturing, it can be seen that the void structure is prone to form some cross structures from the current numerical examples, which will increase difficulty in post-processing. And the optimization result of proposed method is expressed in more detail, which will also increase post-processing work. There are also some hair lines in the optimization structure, such as the

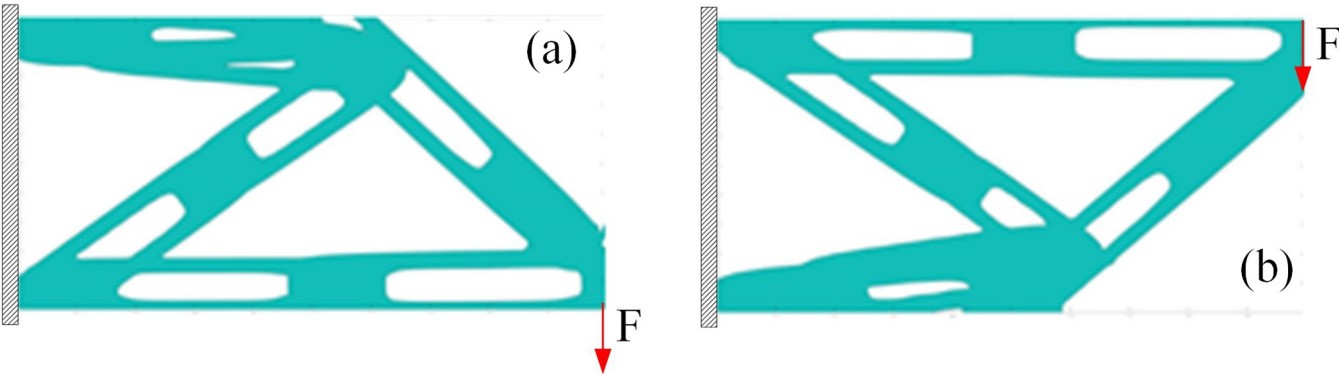

**Fig 16. Topology optimization results of different load positions.** (a) Load on the lower right side. (b) Load on the upper right side.

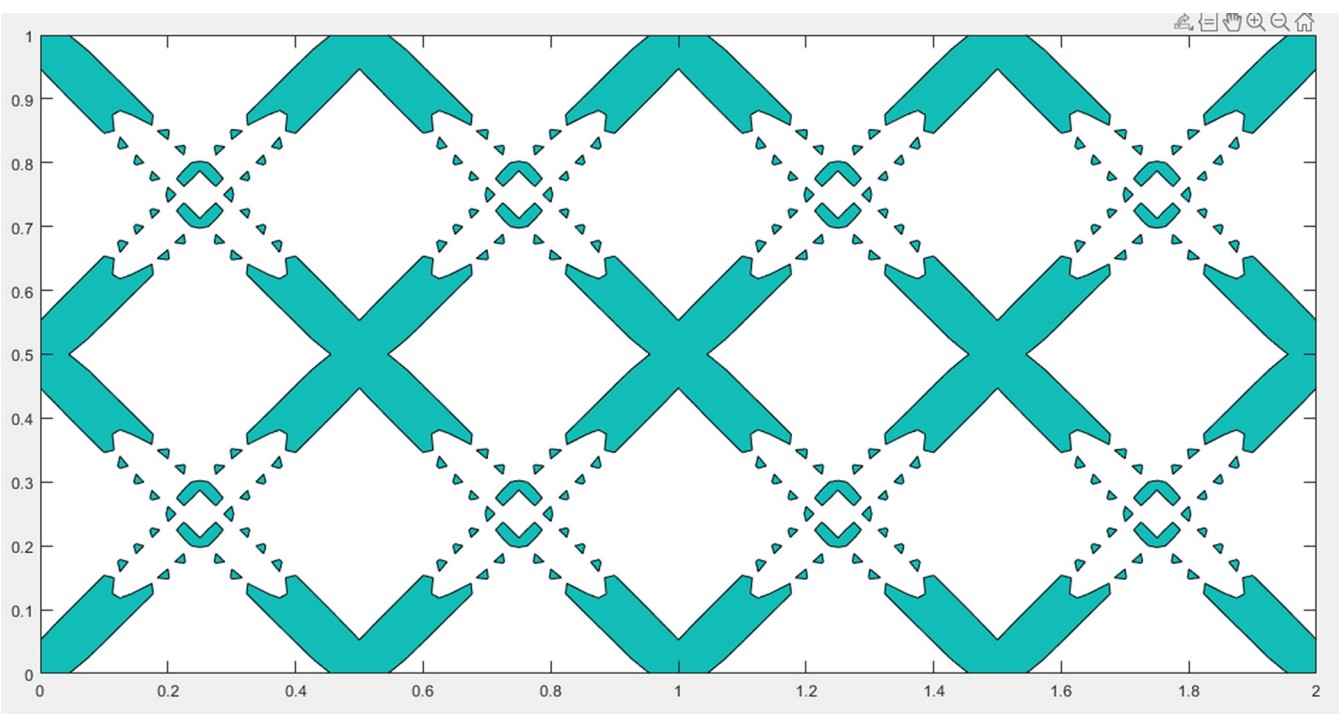

**Fig 17. Schematic diagram of the "islands" phenomenon.**

optimization result with $a = 0.8$ and $b = 0.4$ in Fig 9. Because the boundary of the optimized structure is plotted by the contour lines based on topological function values, it is thought that the emergence of hair lines is due to the fact that the length of component has evolved to be small enough. When the scale factor is larger, this phenomenon is more likely to occur. The hair lines can be attempted to eliminate by constraining the minimum length. In the final optimization result, it should be noted that the proposed method will increase the spatial volume of the final structure under the same volume constraints. In terms of computational efficiency, because each component with void structure need to calculate more topological function values, the computational efficiency of proposed method is reduced. The above limitations need to be noted when using the proposed method and can also be taken as the research directions of the next stage.

## 6 Concluding remakes

Topology optimization is essentially to find the optimal solution by using mathematical methods, so it has a strict mathematical theoretical calculation process, and the feasibility of optimization results is extremely high. At the same time, driven by mathematical optimization algorithms, human intervention is avoided, and many new and more efficient structural forms can be obtained by topology optimization. In the present work, the effects of different void structure scaling factors on MMC topology optimization are discussed. The MMC topology optimization method based on component with void structure mainly has the following characteristics: (1) The component with void structure inherits the advantages of MMC method, with fewer design variables and explicit expression of optimization structure. (2) The variation forms of component are expanded. (3) The scale of internal void structure has a significant impact on the final optimized structure, and the feasible topology structure can be obtained

with suitable scaling factor. (4) As the complexity of component structure increases, more detailed information can be expressed.

It should be noted that during the research of thin-walled component, the "islands" phenomenon has a great influence on the final optimization structure, especially when the "islands" occurs in the initial layout. Therefore, it is necessary to choose the appropriate mesh density and introduce scaling control. The proposed method is mainly to add some macroscopic void structure in the optimization structure. For thin-walled structures, stamping can be used for manufacturing. For non-thin-walled structures, manufacturing may be more difficult, but 3D printing can solve this problem to some extent. Based on the component with void structure, there are many research fields that can be expanded. Some more complex void structures can be set inside the component, such as irregular structure, embedded materials, micro array void structures, etc.

## Supporting information

**S1 File. Matlab codes of MMC topology optimization method based on component with single void structure and double voids structure.**
(ZIP)

## Author Contributions

**Conceptualization:** Zhao Li, Hongyu Xu.

**Data curation:** Hongyu Xu.

**Formal analysis:** Zhao Li.

**Methodology:** Hongyu Xu, Shuai Zhang.

**Software:** Zhao Li.

**Validation:** Shuai Zhang.

**Writing – original draft:** Zhao Li.

**Writing – review & editing:** Hongyu Xu.

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
