## [Decision Letter · Decision Letter 0]

16 Oct 2023

PONE-D-23-29083Moving morphable component (MMC) topology optimization with different void structure scaling factorsPLOS ONE

Dear Dr. Xu,

Thank you for submitting your manuscript to PLOS ONE. After careful consideration, we feel that it has merit but does not fully meet PLOS ONE’s publication criteria as it currently stands. Therefore, we invite you to submit a revised version of the manuscript that addresses the points raised during the review process.

We look forward to receiving your revised manuscript.

Kind regards,

Mario Milazzo

Academic Editor

PLOS ONE

Journal Requirements:

3. Thank you for stating the following financial disclosure: "The financial supports from the Major Science and Technology Project of Henan Province(no.221100240400), and National Natural Science Foundation of China（no.51975244） are gratefully.  acknowledged."

4. Thank you for stating the following in the Acknowledgments Section of your manuscript: "The financial supports from the Major Science and Technology Project of Henan Province(no.221100240400), and National Natural Science Foundation of China（no.51975244） are gratefully acknowledged."

Please remove any funding-related text from the manuscript and let us know how you would like to update your Funding Statement. Currently, your Funding Statement reads as follows: "The financial supports from the Major Science and Technology Project of Henan Province(no.221100240400), and National Natural Science Foundation of China（no.51975244） are gratefully.  acknowledged."

Reviewers' comments:

Reviewer's Responses to Questions

**Comments to the Author**

1. Is the manuscript technically sound, and do the data support the conclusions?

Reviewer #1: Yes

Reviewer #2: Yes

2. Has the statistical analysis been performed appropriately and rigorously? 

Reviewer #1: Yes

Reviewer #2: No

3. Have the authors made all data underlying the findings in their manuscript fully available?

Reviewer #1: Yes

Reviewer #2: No

4. Is the manuscript presented in an intelligible fashion and written in standard English?

Reviewer #1: Yes

Reviewer #2: Yes

5. Review Comments to the Author

Reviewer #1: Overview and general recommendations:

The paper proposes a new description of components for moving morphable component (MMC) topology optimization approach. The method includes internal voids which is determined based on a scaling factor. The paper is well and comprehensively written. However, the issues mentioned below need to be addressed first before it can be considered for publication.

Major comments:

1. The authors stated that the total number of design variables of density method is much greater than that of MMC method. As a result, it has higher optimization efficiency. However, I feel like that for some complicated topologies, for instance, curved beams or curved voids, the MMC method will require a lot of proposed components to form the curvature given their boundaries are straight lines. What are potential solutions to the issue?

2. This work uses a scaling factor to control the internal voids and the factors show a great influence on final results. I am curious that why the authors do not just use two components to represent the material structure and void structure respectively. In other words, why not just use two separate functions to define the boundaries of material region and void region. This will allow more flexibility and less trial-and-error process for scaling factor.

3. The inclusion of void structures is a very interesting and important step for MMC topology optimization. The authors, however, should elaborate more on the limitations of current method. These discussions will further inspire following studies.

4. Can the authors comment on the manufacturing feasibility of proposed method?

Minor comments:

1. The caption of Fig.1 can be more detailed. For instance, what is the meaning of different colors?

2. The authors should provide more examples or explanations about MMC before going into the literature review about different component descriptions. Those questions could be answered in more detailed way:

a. How different components are typically connected in MMC approach, I notice the authors mentioned CMC, but how is general MMC deals with connectivity?

b. No matter what type of components is used, isn’t the design space to be optimized very biased and small given the constrained combination of these shapes?

Reviewer #2: The authors focus on topological optimization technique using moving morphable components, which is a simplified version of topological optimization for reduced DOF and it was published. The authors' main contribution is using different building blocks, including modelled solid, single and double void components using hyperelliptic functions and used flexibility as an objective function to optimize the structures based on loading at a certain point. Results showed that using voids as scaling components in material geometry can be used to obtain final structures. They also showed that single void optimizations are usually better than double voids in terms of computational effectiveness with a tradeoff with less detailed geometries. Many details are missing and there are some issues that needs to be addressed,

1) It is not clear how accurate is the current optimization result, e.g., some validation of the stiffness of the structure in comparing to the other possible structures in finite element simulations.

2) It seems many components are overlapping after optimization (e.g., from fig. 8 to table 2), it will be very useful to use color or other note to highlight the number of componenents placed at the same location.

3) The optimization ideas need to be addressed in greater detail. For example, the importance of the flexibility parameter C on the physical relationship of the structure and the scaling factors(a and b) of length with the constraint on width needs more explanation. Also, it is not clear how p value can affect the geometry of the building block.

4) The optimization of a symmetric initial structure results in asymmetric structures after optimization in some cases. There needs to be a discussion as to why and how it might happen. The authors does not include the boundary conditions (i.e., fixed, loading) in the initial geometry (fig. 8, 11 etc), which should be helpful in understanding the problem

5) Is it possible to come up with a relationship between the mesh size and topological material to avoid having ‘island’ phenomenon during optimization.

6) It is not clear where some features of the optimized structrue comes from. For example, the hair lines in table 2, a=0.8 b=0.4, as these hair lines are not given by the building blocks. To avoid having unusable geometries at the optimized structure there needs to be a discussion on the constraint relationship on the scaling of voids. What constraints, if any, can avoid getting geometry that are unfeasible.

7) Several images (Fig. 1, 2) are copied or reproduced from literature. It is necessary to highlight the source and copyright clarification.

6. PLOS authors have the option to publish the peer review history of their article (what does this mean?). If published, this will include your full peer review and any attached files.

Reviewer #1: No

Reviewer #2: **Yes: **Zhao Qin

---

## [Author Response · Author response to Decision Letter 0]

20 Nov 2023

Dear reviewers and editors,

First of all, thank you very much for your comments, and they are of great help for our research. According to the comments of the reviewers, we have made amendments, supplements and explanations one by one. Please review the details as follows.

The changes made in the manuscript have been marked in red.

Kind regards.

Reviewer #1: 

Overview and general recommendations:

The paper proposes a new description of components for moving morphable component (MMC) topology optimization approach. The method includes internal voids which is determined based on a scaling factor. The paper is well and comprehensively written. However, the issues mentioned below need to be addressed first before it can be considered for publication.

Major comments:

1. The authors stated that the total number of design variables of density method is much greater than that of MMC method. As a result, it has higher optimization efficiency. However, I feel like that for some complicated topologies, for instance, curved beams or curved voids, the MMC method will require a lot of proposed components to form the curvature given their boundaries are straight lines. What are potential solutions to the issue?

Thank you for pointing out this. The reviewer has raised a very meaningful question. The initial components used are usually straight line boundaries. They are difficult to express curved structural features. However, with continuous research on component descriptions, curved components (such as B-spline component, NURBS component) have been applied and have shown good optimization effects in solving curved boundary problems. In the manuscript, references [16-18] use curved components.

2. This work uses a scaling factor to control the internal voids and the factors show a great influence on final results. I am curious that why the authors do not just use two components to represent the material structure and void structure respectively. In other words, why not just use two separate functions to define the boundaries of material region and void region. This will allow more flexibility and less trial-and-error process for scaling factor.

The reviewer's idea is very applicable to many multi-material optimizations based on MMC method. In these cases, two separate functions are used to define the boundaries of strong material region and weak material region (including void region). However, our research focuses on enriching the topology optimization results by changing component description (i.e. increasing void structures), which can provide some references for topology optimization of embedded and perforated structures. 

In our study, the lengths of void structure are associated with the lengths of solid structure, and the evolution process of void structure is constrained by the evolution process of solid structure.

3. The inclusion of void structures is a very interesting and important step for MMC topology optimization. The authors, however, should elaborate more on the limitations of current method. These discussions will further inspire following studies.

Thank you for pointing out this. As suggested by reviewer, the limitations of current method are elaborated more in section 5, and the supplementary content is as follows:

In addition, there are still some limitations of proposed method at present. In manufacturing, it can be seen that the void structure is prone to form some cross structures from the current numerical examples, which will increase difficulty in post-processing. And the optimization result of proposed method is expressed in more detail, which will also increase post-processing work. There are also some hair lines in the optimization structure, such as the optimization result with a=0.8 and b=0.4 in table 2. Because the boundary of the optimized structure is plotted by the contour lines based on topological function values, it is thought that the emergence of hair lines is due to the fact that the length of component has evolved to be small enough. When the scale factor is larger, this phenomenon is more likely to occur. The hair lines can be attempted to eliminate by constraining the minimum length. In the final optimization result, it should be noted that the proposed method will increase the spatial volume of the final structure under the same volume constraints. In terms of computational efficiency, because each component with void structure need to calculate more topological function values, the computational efficiency of proposed method is reduced. The above limitations need to be noted when using the proposed method and can also be taken as the research directions of the next stage.

4. Can the authors comment on the manufacturing feasibility of proposed method?

As suggested by reviewer, the comment on the manufacturing feasibility of proposed method has been added in section 6, and the supplementary content is as follows: 

The proposed method is mainly to add some macroscopic void structure in the optimization structure. For thin-walled structures, stamping can be used for manufacturing. For non-thin-walled structures, manufacturing may be more difficult, but 3D printing can solve this problem to some extent.

Minor comments:

1. The caption of Fig.1 can be more detailed. For instance, what is the meaning of different colors?

Thank you for pointing out this. We have made corresponding modifications to this section. Different colors represent components of different materials, and in order to facilitate understanding, Fig.1 has been modified and explanations have been added. The supplementary content is as follows:

First, topology optimization (TO) model should be created, mainly referring to the problem formulation, including design variable, objective function, and constraint. Then numerical implementation method should be established. In MMC framework, the number, type, and distribution of initial components should be determined firstly. A set of given initial layout components are projected onto the background mesh to obtain the information of mesh nodes in the projection area. The finite element analysis is used to solve the objective function. The design variables are updated based on constraint conditions and optimization algorithms, and the geometric information of the components will be changed. The optimal solution is obtained according to the convergence conditions. The implementation process of MMC method is shown in Fig.1.

2. The authors should provide more examples or explanations about MMC before going into the literature review about different component descriptions. Those questions could be answered in more detailed way:

a. How different components are typically connected in MMC approach, I notice the authors mentioned CMC, but how is general MMC deals with connectivity?

b. No matter what type of components is used, isn’t the design space to be optimized very biased and small given the constrained combination of these shapes?

Thank you for pointing out this. As suggested by reviewer, we have revised the related contents.

(1) The explanations about MMC have been added in the introduction, and the supplementary content is as follows:

Compared with traditional topology optimization methods where structural topologies are represented either by element densities (in density method) or by nodal values of a level set function (in level set method), in MMC method, a set of moving morphable components are adopted as basic building blocks of topology optimization. These components are allowed to move, deform, overlap and merge in the design domain freely, and structural topology can be obtained by changing the positions, inclined angles, lengths and the layout of these components. Moreover, these components comprise explicit geometric parameter information, which can be directly evolved to optimize structural topology.

(2) Component connection is an interesting and important research direction in MMC method. At present, there is relatively little research on component connection methods. Usually, component connection positions can be reserved by setting undersigned area at both ends of the component. For specific connection methods, thin-walled structures can be directly stamped into shape, and welding or riveting can be used for other structures. 

(3) In MMC method, the layout of the initial components and the shape of the components are also important. Theoretically speaking, using more components in the initial design has the effects of enlarging the design space, enhancing the possibility for finding an appropriate optimized design and alleviating the dependence of initial design. However, increasing the number of components usually implies extra computational cost, and it is difficult to predict an appropriate initial components’ distribution, especially for complex problems. Therefore, it is critical to find a reasonable initial component layout in order to avoid the failure of MMC topology optimization, but it is still challenging.

 

Reviewer #2: 

The authors focus on topological optimization technique using moving morphable components, which is a simplified version of topological optimization for reduced DOF and it was published. The authors' main contribution is using different building blocks, including modelled solid, single and double void components using hyperelliptic functions and used flexibility as an objective function to optimize the structures based on loading at a certain point. Results showed that using voids as scaling components in material geometry can be used to obtain final structures. They also showed that single void optimizations are usually better than double voids in terms of computational effectiveness with a tradeoff with less detailed geometries. Many details are missing and there are some issues that needs to be addressed,

1) It is not clear how accurate is the current optimization result, e.g., some validation of the stiffness of the structure in comparing to the other possible structures in finite element simulations.

Thank you for pointing out this. The comparison between the current optimization result and the solid component optimization result has been added in section 5.3.2, and the supplementary content is as follows:

The void structure makes the component thinner, which has significant impact on the final optimization result, including the topology and minimum flexibility value. Some void structures even cause the optimization results without practical significance. However, compared with the optimization results of solid component, as shown in Fig.12, the void structure with an appropriate scaling factor can achieve the similar accuracy.

2) It seems many components are overlapping after optimization (e.g., from fig. 8 to table 2), it will be very useful to use color or other note to highlight the number of componenents placed at the same location.

As suggested by reviewer, the supplementary content is as follows:

The layout of the components can be identified in Fig.9. It can be clearly seen that the lengths of components have changed and the components are overlapped at the same location.

3) The optimization ideas need to be addressed in greater detail. For example, the importance of the flexibility parameter C on the physical relationship of the structure and the scaling factors(a and b) of length with the constraint on width needs more explanation. Also, it is not clear how p value can affect the geometry of the building block.

Thank you for pointing out this. 

（1）The supplementary content is as follows:

Structural flexibility C represents the deformation energy of solid structure, which is the sum of deformation energy caused by body and surface forces. Based on finite element analysis ideas, structural flexibility C is also equal to the sum of strain energy of elements. In MATLAB program implementation, finite element analysis is used to calculate the structural flexibility. And in MMC framework, the premise of finite element analysis is to obtain solid element information based on nodes’ topological function values and Heaviside function. The stiffness matrix is solved by ersatz material model. In our proposed method, the role of the scaling factors (a and b) is mainly worked in program implementation. The voids generated by the scaling factors (a and b) are used to remove the internal nodes of the voids. The removed internal nodes of the voids will no longer undergo finite element analysis, and the removal of voids is achieved by the Min function in this paper.

（2）The hyperelliptic equation is used to describe component in this paper. The general form of hyperelliptic equation is: （x/a）p+(y/b)p=1

The p value can affect the geometry of the building block. 

When 0<p<1, the geometric of hyperellipse is similar to a four cornered star curve with four sides concave outward.

When p=1, the geometric of hyperellipse is a diamond. 

When 1<p<2, the geometric of hyperellipse is similar to a diamond, with four sides convex outward.

When p=2, the geometric of hyperellipse is an ellipse.

When p>2, the geometric of hyperellipse is a rounded rectangle.

For the description of component in MMC method, many numerical examples have confirmed that good optimization results can be achieved when p=6.

4) The optimization of a symmetric initial structure results in asymmetric structures after optimization in some cases. There needs to be a discussion as to why and how it might happen. The authors does not include the boundary conditions (i.e., fixed, loading) in the initial geometry (fig. 8, 11 etc), which should be helpful in understanding the problem

Thank you for pointing out this. We agree with this comment.

（1）The supplementary content is as follows:

It should be noted that the evolution of components is achieved by the method of moving asymptotes (MMA) optimizer based on gradient information, and the initial symmetrical layout of components is only convenient for setting. When a stable load-transferring path is established, it is the truly effective "initial layout". 

For initial layout in MMC method, theoretically speaking, using more components in the initial design has the effect of enlarging the design space, enhancing the possibility for finding an appropriate optimized design and alleviating the dependence of initial design. However, increasing the number of components usually implies extra computational cost, and it is difficult to predict an appropriate initial components’ distribution, especially for complex problems. Therefore, it is critical to find a reasonable initial component layout in order to avoid the failure of MMC topology optimization, but it is still challenging. 

（2）The boundary conditions (fixed position and loading position) have been marked in the initial geometry (Fig. 8, Fig.9, and Fig.11).

5) Is it possible to come up with a relationship between the mesh size and topological material to avoid having ‘island’ phenomenon during optimization.

From the current numerical analysis, it can be seen that fine mesh can avoid the "island" in the initial layout of components, which is very important for establishing an effective load-transferring path. However，in the process of optimization, the lengths of component will change, and some components may become too small to be captured by the mesh (relatively speaking, it can also be due to the excessive mesh division scale). These structures will not work in finite element analysis, but will still display in the final optimization result, independent of the main structure. 

6) It is not clear where some features of the optimized structrue comes from. For example, the hair lines in table 2, a=0.8 b=0.4, as these hair lines are not given by the building blocks. To avoid having unusable geometries at the optimized structure there needs to be a discussion on the constraint relationship on the scaling of voids. What constraints, if any, can avoid getting geometry that are unfeasible.

As suggested by reviewer, we have revised the related contents, and the supplementary content is as follows:

There are also some hair lines occur in the optimization structure, such as the optimization result with a=0.8 and b=0.4 in table 2. Because the boundary of the optimized structure is plotted by the contour lines based on topological function values, we think the emergence of hair lines is due to the fact that the length of component evolves to be small enough. When the scale factor is larger, this phenomenon is more likely to occur. The hair lines can be attempted to eliminate by constraining the minimum length.

7) Several images (Fig. 1, 2) are copied or reproduced from literature. It is necessary to highlight the source and copyright clarification.

As suggested by reviewer, the sources of the images have been added with relevant references.

---

## [Decision Letter · Decision Letter 1]

12 Dec 2023

Moving morphable component (MMC) topology optimization with different void structure scaling factors

PONE-D-23-29083R1

Dear Dr. Xu,

We’re pleased to inform you that your manuscript has been judged scientifically suitable for publication and will be formally accepted for publication once it meets all outstanding technical requirements.

Kind regards,

Mario Milazzo

Academic Editor

PLOS ONE

Additional Editor Comments (optional):

Reviewers' comments:

Reviewer's Responses to Questions

**Comments to the Author**

1. If the authors have adequately addressed your comments raised in a previous round of review and you feel that this manuscript is now acceptable for publication, you may indicate that here to bypass the “Comments to the Author” section, enter your conflict of interest statement in the “Confidential to Editor” section, and submit your "Accept" recommendation.

Reviewer #1: All comments have been addressed

Reviewer #2: All comments have been addressed

2. Is the manuscript technically sound, and do the data support the conclusions?

Reviewer #1: Yes

Reviewer #2: Yes

3. Has the statistical analysis been performed appropriately and rigorously? 

Reviewer #1: Yes

Reviewer #2: Yes

4. Have the authors made all data underlying the findings in their manuscript fully available?

Reviewer #1: Yes

Reviewer #2: Yes

5. Is the manuscript presented in an intelligible fashion and written in standard English?

Reviewer #1: Yes

Reviewer #2: Yes

6. Review Comments to the Author

Reviewer #1: All comments have been addressed by the authors in the revision, thus I recommend the paper for publication.

Reviewer #2: (No Response)

7. PLOS authors have the option to publish the peer review history of their article (what does this mean?). If published, this will include your full peer review and any attached files.

Reviewer #1: No

Reviewer #2: No

---

## [Editor Report · Acceptance letter]

19 Dec 2023

PONE-D-23-29083R1 

PLOS ONE

Dear Dr. Xu, 

I'm pleased to inform you that your manuscript has been deemed suitable for publication in PLOS ONE. Congratulations! Your manuscript is now being handed over to our production team.

Kind regards, 

on behalf of

Dr. Mario Milazzo 

Academic Editor

PLOS ONE